# Reforestation Based on Mono-Plantation of Fast-Growing Tree Species Make It Difficult to Maintain (High) Soil Water Content in Tropics, a Case Study in Hainan Island, China

**Wenjun Hong** [1,†]**, Jindian Yang** [1,†]**, Jinhuan Luo** [1,†]**, Kai Jiang** [2,3]**, Junze Xu** [1] **and Hui Zhang** [2,3,]*

1    Sanya Academy of Forestry, Sanya 572000, China; hongwenjun0827@126.com (W.H.);
     zhanghui1985052785@126.com (J.Y.); ljh3779@163.com (J.L.); zhanghuitianxia@163.com (J.X.)
2    College of Forestry/Wuzhishan National Long Term Forest Ecosystem Monitoring Research Station,
     Hainan University, Haikou 570228, China; y_feng000@163.com
3    Key Laboratory of Genetics and Germplasm Innovation of Tropical Special Forest Trees and Ornamental
     Plants (Hainan University), Ministry of Education, College of Forestry, Hainan University,
     Haikou 570228, China
*    Correspondence: 993781@hainu.edu.cn
†    These three authors contributed equally.

**Abstract:** Reforestation has been assumed as a natural solution to recover soil water content, thereby increasing freshwater supply. Mono-plantation of fast-growing species is the first step for performing reforestation to prevent frequent and heavy rain-induced landslide in tropics. However, fast-growing species may have negative hydraulic response to seasonal drought to maintain high growth rate and, thus, may make it difficult for reforestation in tropics to recover soil water content. We tested this hypothesis in a setting involving (a) a reforestation project, which mono-planted eight fast-growing tree species to successfully restore a 0.2-km$^2$ extremely degraded tropical rainforest, and (b) its adjacent undisturbed tropical rainforest in Sanya City, Hainan, China. We found that, for maintaining invariably high growth rates across wet to dry seasons, the eight mono-planted fast-growing tree species had comparable transpiration rates and very high soil water uptake, which in turn led to a large (3 times) reduction in soil water content from the wet to dry seasons in this reforested area. Moreover, soil water content for the adjacent undisturbed tropical rainforest was much higher (1.5 to 5 times) than that for the reforested area in both wet and dry seasons. Thus, the invariably very high water demand from the wet to dry seasons for the mono-planted fast-growing species possesses difficulty in the recovery of soil water content. We suggest, in the next step, to mix many native-species along with the currently planted fast-growing nonnative species in this reforestation project to recover soil water content.

**Keywords:** deforestation; freshwater scarcity; hydraulic response to seasonal drought; limited leaf water supply; recovery of soil water content; tropical rainforest reforestation

---

## 1. Introduction

Human beings face a freshwater scarcity problem on account of the steadily increasing freshwater demand [1]. Currently, soil water content is one of the main freshwater resources [2], and globally, forests play a key role in maintaining them [3]. Historic human disturbance (e.g., ore mining and unreasonable agricultural use) have resulted in very high deforestation and degradation in tropical rainforest worldwide, which in turn has led to a large amount of global freshwater loss [4–7]. Thus,

a number of reforestation projects have been performed worldwide to alleviate global water loss [8–11]. However, relatively few studies have evaluated the influence of large spatial or temporal reforestation projects on soil water content, especially in the humid subtropics and the tropics [12].

Forest evapotranspiration and enhanced soil infiltration can increase rainfall and soil water content [13], which are two main resources of freshwater [2]. Since the tropics would potentially witness high amounts of deforestation in the near future [14], this in turn may result in a large amount of freshwater loss. Indeed, many studies have found that the high deforestation in tropics have led to large scale freshwater loss [15–19]. A previous meta-analysis has found that reforestation-induced changes of landscape composition and configuration may be an effective way to increase freshwater supply in the tropics [20]. However, nearly no study has evaluated the influence of large spatial or temporal reforestation projects on soil water content in humid tropics [8].

Frequent typhoon and heavy rainfall during the monsoon season could easily cause landslides and tree lodging in tropical rainforest [16–19,21]. This make the reforestation of large areas of highly degraded tropical rainforests very difficult. Performing reforestation while using mono-plantation of nonnative fast-growing tree species with high survival rate could help in preventing landslide [22–24]. However, mono-plantation of nonnative fast-growing species may also lead to fast reductions in soil water content. That is because forest transpiration and its enhanced soil water infiltration are the key determinants of the final soil water content [9,10]. As shown in Figure 1a, when precipitation is absorbed into the soil, forest transpiration acts as a pump, as one part of the water is absorbed and finally lost to the atmosphere. Soil water infiltration, however, is like a sponge that permeates the rest of the precipitation and finally keeps them in the deep soil layers. Fast-growing trees usually have high transpiration [11], which may further trigger soil water uptake, reduce soil water infiltration [12], and thereby result in a low soil water content (Figure 1b). Thus, it may be difficult to recover soil water content by reforestation using mono-planted fast-growing species.

To investigate this, since 2016, we have performed a reforestation project in Baopoling mountains (BPL) in Sanya, Hainan island, China, which involves separately mono-planting eight fast-growing tree species to restore a 0.2 km$^2$ highly degraded tropical forest after ore mining. Specifically, we used information on the topographic and soil environments in an adjacent undisturbed tropical rainforest as a reference for this reforestation project. By this way, comparing the differences in transpiration rates between these eight fast-growing tree species and eight dominant tree species in the adjacent undisturbed tropical rainforest could reveal whether reforestation based on monoculture of fast-growing tree species might indeed recover soil water content.

Tree transpiration is also determined by some key tree hydraulic responses including photosynthesis rate, stomatal conductance, leaf hydraulic conductivity, and drought stress tolerance [13,25–29]. Functional traits (maximum photosynthesis rate, transpiration rate stomatal conductance, leaf hydraulic conductivity, and leaf turgor loss point) can directly capture these hydraulic responses [30,31]. It has been found that there is a seasonal drought in BPL [32], which should result in different hydraulic responses by the fast-growing "introduced" tree species and the slow-growing dominant tree species in the wet and the dry seasons. Thus, here, we compared the differences in these functional traits between the introduced fast-growing tree species and the slow-growing dominant tree species in the adjacent undisturbed tropical rainforest. We also tested how soil water content in both the reforestation project and the undisturbed tropical rainforest vary from the wet to dry seasons. We hypothesized that fast-growing species would develop negative hydraulic responses (e.g., having much higher transpiration and leaf hydraulic conductivity but lower drought stress tolerance than the slow-growing dominant tree species) between the wet and the dry seasons, thereby limiting recovery of soil water content during this reforestation project.

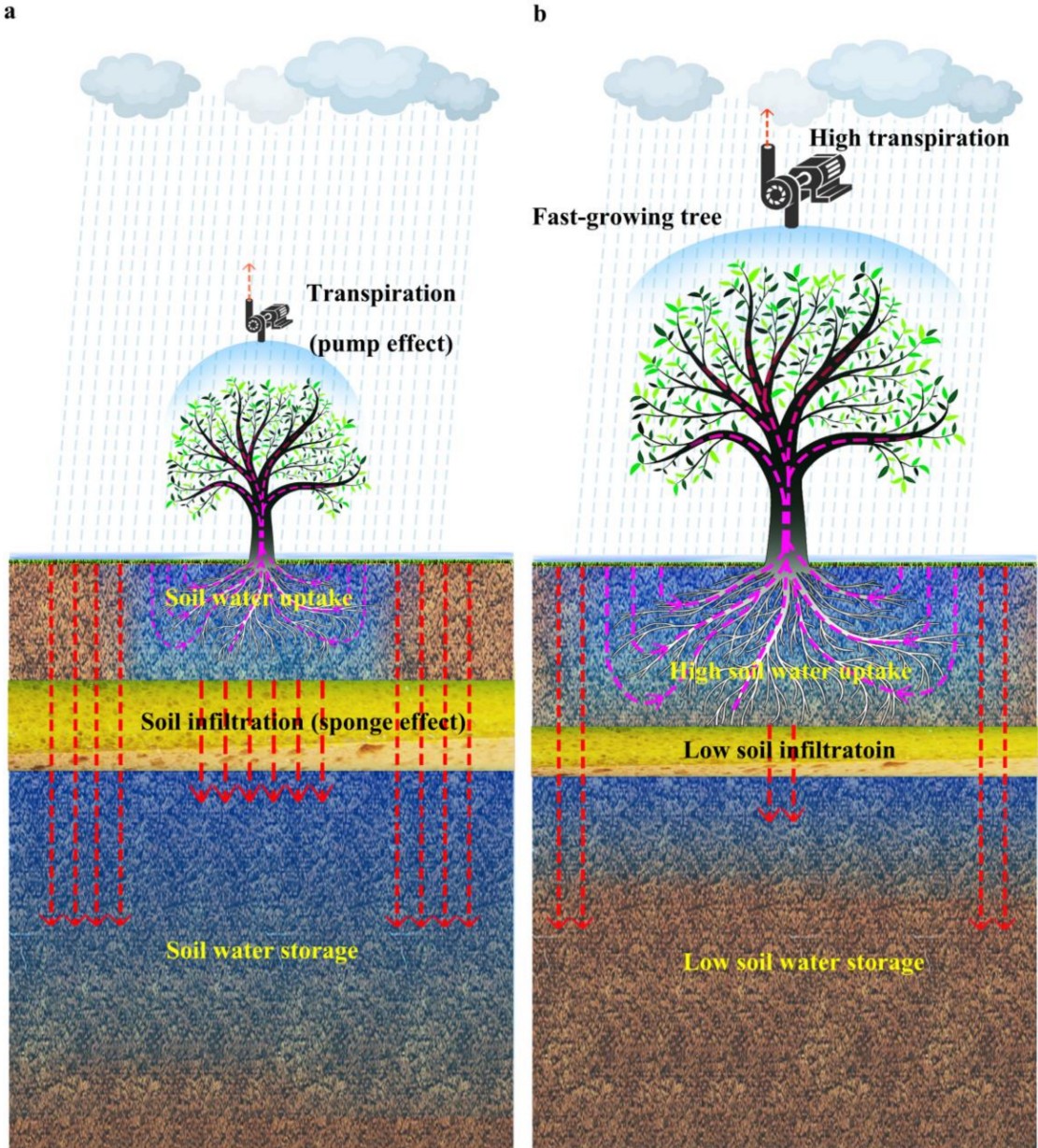

**Figure 1.** Hypothesizing (**a**) the roles of tree transpiration and soil infiltration, and (**b**) the influences of fast-growing tree species on soil water content.

## 2. Materials and Methods

### 2.1. Study Sites

Our study site was located in the Baopoling mountain, which is a limestone mountain in Sanya City, Hainan China (BPL, 109°51′01″ E, 18°31′99″ N; Figure 1). It has a tropical monsoon oceanic climate with a mean annual temperature of 28 °C. The average annual precipitation on the island is 1500 mm, and most (91%) of the precipitation occurs in the wet season (June to October) [32]. The inhabitants of the village near BPL get their water supply from the nearby pond and the water works in Sanya city. Additionally, the occurrence of a major cement factory (Huasheng cement factory, China) near BPL exponentially increases the water demand so that the city sometimes gets very limited water supply in the dry season. The typical vegetation of the BPL is a species-rich tropical monsoon broad-leaf forest.

Due to 20 years of limestone mining associated with the cement industry, this 0.2 km$^2$ highly degraded tropical forest now consists merely of bare rocks that do not support plant life (Figure 2). Areas of the BPL outside of this 0.2 km$^2$ degraded area have been significantly disturbed and, therefore, have remained as a species-rich tropical rainforest (Figure 2). In May 2016, we used the adjacent undisturbed forest as a reference to perform a reforestation project in BPL with the aim to recover soil water content and vegetation cover of BPL. The slope and deep soil layers of the undisturbed forest area were used as a reference to reconstruct slope and soil layers for the reforested area. Then, refilling of the area was performed with the help of the soil from the undisturbed tropical rainforest areas to monoculture seedlings (3 m height and 2 cm diameter at breast height (DBH)) of eight fast-growing tree species: *Terminalia neotaliala*, *Bombax malabarica*, *Cleistanthussumatranus*, *Ficusmicrocarpa*, *Muntingiacolabura*, *Acacia mangium*, *Leucaena glauca* and *Bougainvillea spectabilis*. Seedlings of these eight fast-growing species were purchased commercially. These species are known to be fast-growing and have high survival rates within the study region. Therefore, we reasoned that these eight species should have high potential to prevent landslides during frequent typhoon and heavy rains. These eight species were separately monocultured from the top to the bottom of BPL (Figure 2), and planting density for each of the species was maintained at 100 stems per hectare. The restoration project was finished at the end of the year 2016. In 2019, thirty plots, each of 20 × 20 m$^2$ (an area of 400 m$^2$ for each plot) that were at least 100–300 m apart from one another, were randomly sampled across the adjacent undisturbed old-growth forest. Within each plot, all freestanding trees with diameter of ≥1 cm at breast height (DBH) were measured and identified to species. *We finally found 80 tree species in the undisturbed old-growth forest, and we selected the 8 tree species (200–300 stems per hectare) Brideliatomentosa, Radermacherafrondosa, Lepisanthesrubiginosa, Rhaphiolepisindica, Pterospermumheterophyllum, Fissistigmaoldhamii, Psychotria rubra, and Cudraniacochinchinensis as our candidate dominant slow-growing tree species.*

*2.2. Sampling*

We selected two sites (A and B) in the reforested and the undisturbed areas, respectively, in BPL (Figure 2). In the peak of the wet season (August) in 2019, we sampled 20 fully expanded, healthy leaves from the same five independent individuals for each of the eight fast-growing species and the eight dominant slow-growing species found in the surrounding undisturbed region. Resampling was performed in the dry season (February) in 2020. Leaf samples were used to measure five hydraulic traits: transpiration rate (TR; $\mu$mol m$^{-2}$ s$^{-1}$), maximum photosynthesis rate (A$_{mass}$; $\mu$mol m$^{-2}$ s$^{-1}$), stomatal conductance (SC; mmol m$^{-2}$ s$^{-1}$), leaf hydraulic conductivity (LHC; mmol m$^{-2}$ s$^{-1}$ MPa$^{-1}$), and leaf turgor loss point (TLP; Mpa). Detailed descriptions of the trait measurements are provided in the Supplementary Materials. We also collected 30 soil samples at a depth of 0–100 cm at site A and site B, respectively, to measure soil water content (mg kg$^{-1}$) gravimetrically. Every soil sample was homogenized for the whole depth (0–100 cm). Resampling was performed again in the dry season (February) in 2020.

*2.3. Statistics Methods*

First, we used a nonparametric test (generalized linear mixed effect model with Poisson error family) ($p < 0.05$) to test whether there were differences in transpiration-related functional traits (transpiration rate, photosynthesis rate, stomatal conductance, leaf hydraulic conductivity, and leaf turgor loss point) between the eight nonnative and the eight native tropical tree species. We also compared the differences in soil water content between site A (reforestation area) and site B (undisturbed area). A generalized linear mixed effect model with Poisson error family was carried out by function *"glmer"* in R package *"lme4"*.

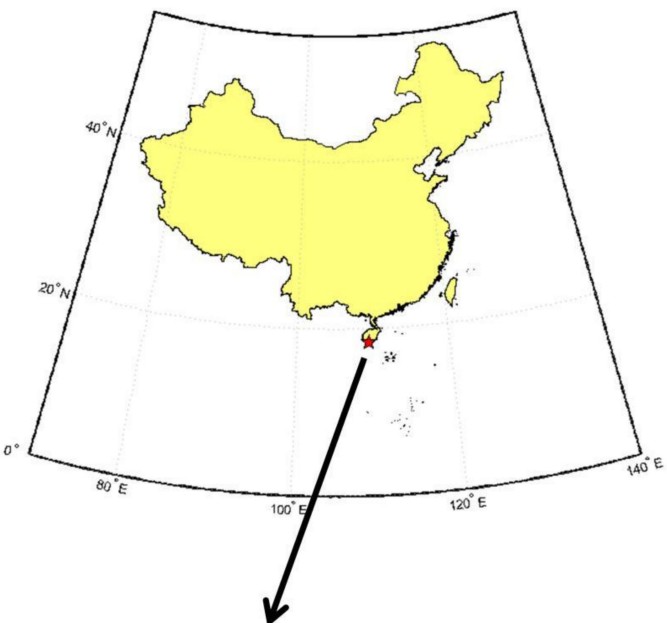

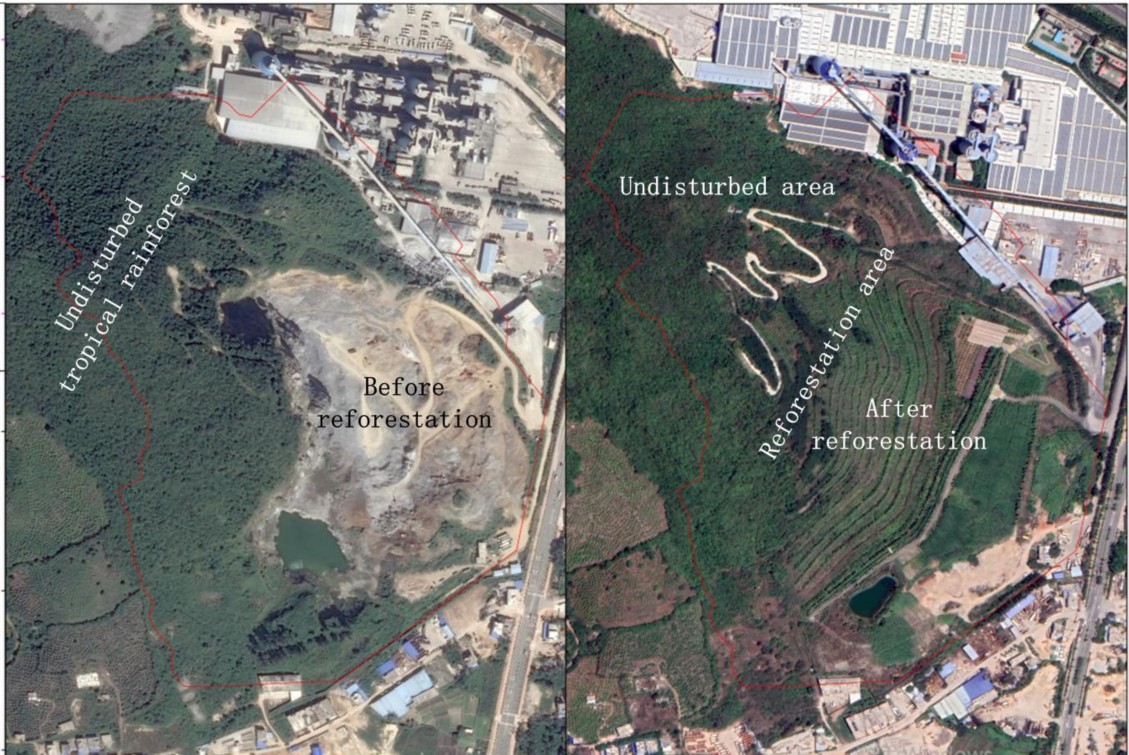

**Figure 2.** The map of the study site (Baopoling mountain) and the landscape of the 0.2 km$^2$ highly degraded area before and after reforestation: leaf samples for the eight nonnative and the native tree species, and soil samples were collected from both site A (reforestation area) and site B (undisturbed area).

## 3. Results

The entire project was completed in 2017, and in the past two years, both typhoon and heavy rains during the wet seasons have never caused landslide and tree lodgings on the reforested site. The mean precipitation per month in the wet and dry seasons were 1380 mm and 182 mm, respectively (Figure S1 in the Supplementary Materials). The native dominant tree species in the dry season had significantly

much lower (from 1/5th to 1/2th) transpiration rate, photosynthesis rate, stomatal conductance, lead hydraulic conductivity, and leaf turgor loss point than those in the wet season (Figure 3; $p < 0.05$, generalized linear mixed effect model with Poisson error family). In contrast, compared to the wet season, nonnative fast-growing species had significantly higher (5 times) transpiration rate and leaf hydraulic conductivity, but photosynthesis rate, stomatal conductance and leaf turgor loss point were not significantly different (Figure 3). In the wet season, nonnative species have significantly higher (from 2 to 4 times) transpiration rate, photosynthesis rate, stomatal conductance, and leaf hydraulic conductivity but comparable leaf turgor loss point, compared to those of the native species (Figure 4). During the wet season, all five traits for nonnative fast-growing tree species were much higher (6 times) than those for native dominant tree species (Figure 4).

Soil water content slightly decreased from the wet to the dry seasons, in the undisturbed ecosystem, whereas soil water content in the wet season was 2.1 times the dry season in the reforested area (Figure 5). Moreover, soil water content for the undisturbed ecosystem is much higher (1.5 to 2.5 times) than those for the reforestation project in both the seasons (Figure 5).

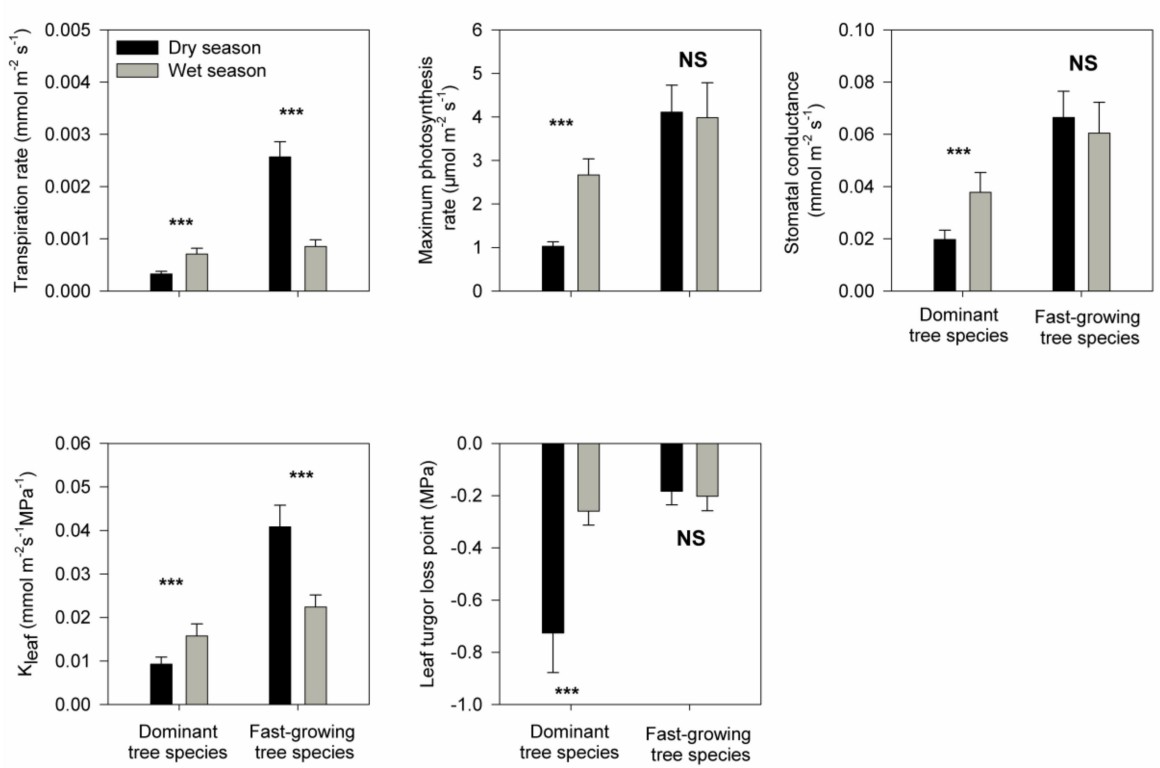

**Figure 3.** Differences in each of the five hydraulic traits (transpiration rate (TR), maximum photosynthesis rate (Amass), stomatal conductance (SC), leaf hydraulic conductivity (LHC), and leaf turgor loss point (TLP)) between the wet and the dry seasons for the fast-growing species used for reforestation as well as the dominant slow-growing species in the adjacent undisturbed tropical rain forest. *** indicates $p < 0.05$ and NS (nonsignificant) indicates $p > 0.05$ based on a generalized linear mixed effect model with Poisson error family. Bars indicate the mean values, and error bars denote standard errors.

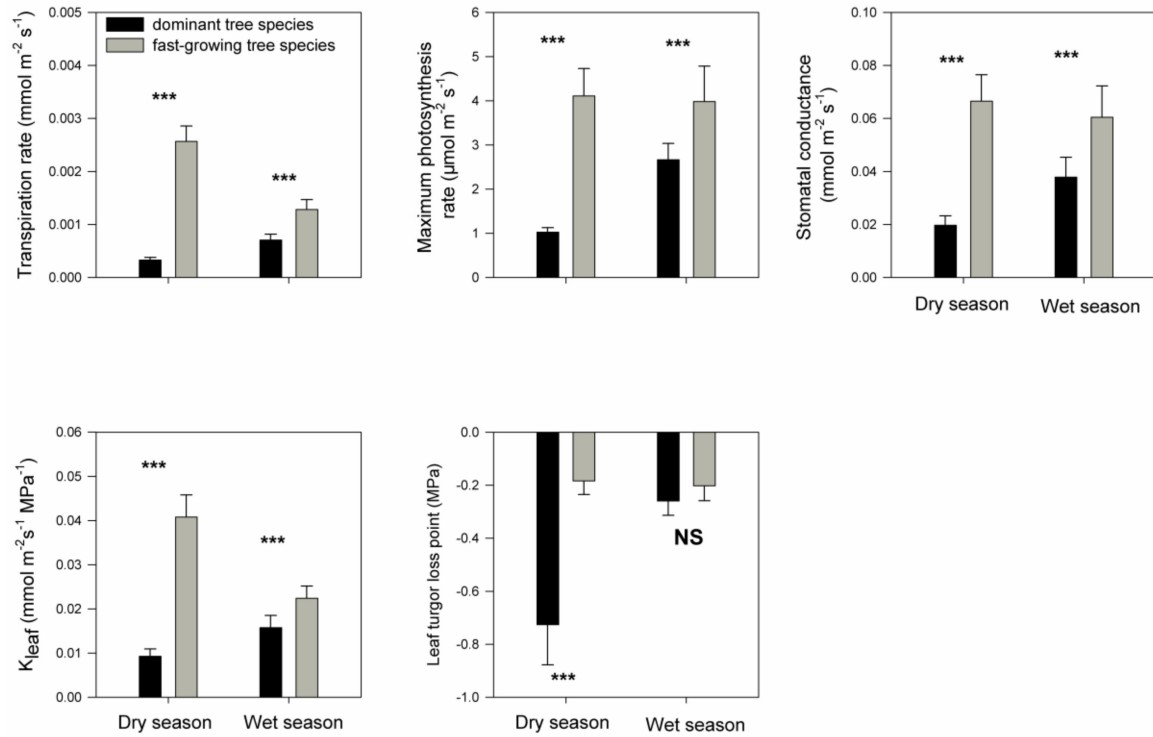

**Figure 4.** Differences in each of the five hydraulic traits (transpiration rate (TR), maximum photosynthesis rate (Amass), stomatal conductance (SC), leaf hydraulic conductivity (LHC), and leaf turgor loss point (TLP)) between the fast-growing species used for reforestation and the dominant slow-growing species in the adjacent undisturbed tropical rain forest in the dry and the wet seasons, respectively. *** indicates $p < 0.05$ and NS (nonsignificant) indicates $p > 0.05$ based on a generalized linear mixed effect model with Poisson error family. Bars indicate the mean values, and error bars denote standard errors.

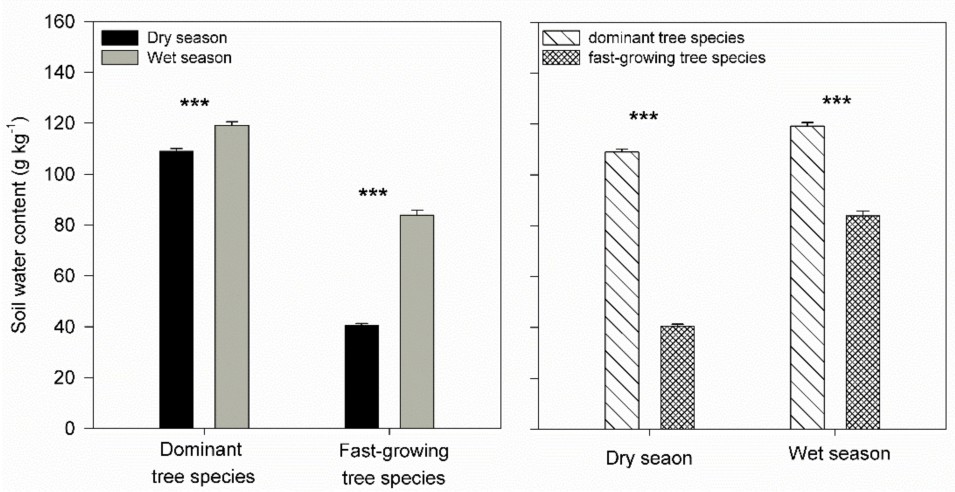

**Figure 5.** Differences in the soil water contents of reforested area and its adjacent undisturbed tropical rainforest ecosystem in the wet and the dry seasons. *** indicates $p < 0.05$ and NS (nonsignificant) indicates $p > 0.05$ based on generalized linear mixed effect model with Poisson error family. Bars indicate the mean values, and error bars denote standard errors.

## 4. Discussion

Using a large area of reforestation project, we evaluated whether reforestation could recover soil water content in humid tropic. The reforestation in this project was conducted by mono-planting

eight nonnative fast-growing tree species to recover a highly degraded tropical rainforest ecosystem; We hypothesized that the very high photosynthesis water demand across wet and dry seasons by the mono-planted nonnative species might result in equally high soil water uptake thereby leading to a very low soil water content. Therefore, in the current stage of this reforestation project, it cannot help recover soil water content.

We found that the mean precipitation in the dry season is merely 1/3 of those in the wet season. Moreover, we found that the eight native species had much lower leaf turgor loss point in the dry season than that in the wet season. Lower leaf turgor loss point is usually observed when a leaf cannot get enough water supply [33]. These results indicated that all the native trees have very limited water supply in the dry season. Limited water supply usually would constrain the photosynthesis rate [34], which in turn could decrease the transpiration rate, stomatal conductance, and leaf hydraulic conductivity to decrease the photosynthesis water demand [25,27,35]. Indeed, in the dry season, the eight native tree species had lower transpiration rate, photosynthesis rate, stomatal conductance, leaf hydraulic conductivity, and leaf turgor loss point than those in the wet season. In addition, lower TLP also indicated higher drought stress tolerance [36]. As a result, for adapting to the limited water supply in the dry season, native species were inclined towards having low photosynthesis water need and high drought stress tolerance, which in turn resulted in lower soil water uptake but higher soil infiltration, thereby maintaining a very high soil water content. Consistent with this hypothesis, we observed a slightly decreased soil water content from the wet to the dry seasons. Thus, native tree species' appropriated hydraulic responses to seasonal drought could result in a slight decrease in the soil water content from the wet to the dry seasons.

Fast growth usually requires a high photosynthesis rate [30,31,37], which in turn leads to higher transpiration rate, stomatal conductance, and leaf hydraulic conductivity [13,25–28]. Indeed, we found that the eight nonnative species had much higher transpiration, photosynthesis rate, stomatal conductance, and leaf hydraulic conductivity than those for native species in both seasons. This demonstrated that these nonnative species require high photosynthesis water demand to maintain a very high maximum photosynthesis rate, which in turn may lead to lower soil water content [34]. Indeed, soil water content, as indicated in the reforestation project, is much lower than those for the undisturbed tropical rainforest ecosystem. Thus, use of mono-plantation of fast-growing tree species would lead to very low soil water content. This has also been observed in rubber and *Eucalyptus* monocultures [29,38]. In contrast, native species could facilitate higher soil water content than the nonnative species, which has also been observed in other reforestation projects [39].

We did not find significant differences in leaf turgor loss point (TLP) between nonnative and native species in the wet season, whereas nonnative species had much higher TLP than that for native-species in the dry season. Moreover, TLP for nonnative species did not vary from the wet to dry seasons. These results indicated that, in the wet season, both the nonnative and the native species could have enough leaf water supply, which leads to comparable TLP between the nonnative species and the native species. In contrast, a very limited leaf water supply might have helped the native species in their higher drought stress tolerance but native species had enough leaf water supply. It is very surprising that the photosynthesis rate for nonnative species are much higher (5 times) than those for native species in the dry season and that the photosynthesis rate for nonnative species did not vary between the wet and the dry seasons. Thus, the nonnative species should require much higher leaf water supply than the native species in the dry season: this appears impossible due to the limited water supply in the dry season. One possibility is that nonnative species tend to have higher transpiration and leaf hydraulic conductivity that absorb a large amount of soil water. This could help the plants maintain leaf water supply, thereby having higher photosynthesis rates in the dry season. Indeed, fast-growing species had high transpiration rates and leaf hydraulic conductivities. Moreover, soil water content in the reforestation area in the dry season was considerably less than in the wet season and it was also significantly less than that for the undisturbed ecosystem in the dry season. As a result, very high

water demands for the mono-planted non-native species might have resulted in very high soil water uptake, which in turn would lead to a very low soil water content.

## 5. Conclusions

Here, we found that the very high water demand by the nonnative species across the whole year in this reforestation project raises difficulties for retaining high soil water content. The adjacent native species' low water needs and the high drought stress tolerance could help restore high soil water content in the dry season. Although reforestation based on mono-planting fast-growing tree species does not seems to maintain high soil water content, we still suggest fast-growing tree species to be the first step in performing reforestation of degraded tropical forests, as this could help prevent landslides and tree lodgings that occur due to frequent typhoons and heavy rains in the tropics. Moreover, plantation of fast-growing tree species could also increase microbial diversity and abundances [40]. In addition, fast-growing tree species dominated tropical rainforest in early successions, and the slow-growing tree species will gradually replace the fast-growing tree species in late succession, when soil nutrient and water are not enough to sustain the growth and survival of fast-growing tree species [41]. Thus, plantation of fast-growing tree species could also facilitate the recovery of the original tropical rainforest in BPL. However, for gradually recovering soil water content in BPL, we suggest mixing adjacent native tree species with these mono-plantation. However, the following studies should be performed in the future when planning to mix the adjacent dominant tree species with fast-growing tree species: (1) after what time should the native tree species be replanted together with fast-growing species? (2) Once native trees have been planted with fast-growing trees in the reforestation project, will and by when should the soil water fully recover to the levels of undisturbed forests.

**Supplementary Materials:** The following are available online at http://www.mdpi.com/2073-4441/12/11/3077/s1, Figure S1: The variations of mean precipitation in wet (June to October) and dry seasons (November to May) based on precipitation record per month from the local weather bureau in Sanya City, China, Text S1: Methods for functional trait measurements.

**Author Contributions:** Conceptualization, W.H., J.L., H.Z., and J.Y.; methodology, W.H., J.L., H.Z., and J.Y.; software, J.X. and K.J.; formal analysis, H.Z. and J.Y.; validation: W.H., J.L., H.Z., and J.Y.; investigation, J.L.; data curation, W.H., J.L., H.Z., and J.Y.; writing—original draft preparation, W.H., J.L., H.Z., and J.Y.; writing—review and editing, W.H., J.L., H.Z., and J.Y.; visualization, H.Z.; supervision, H.Z.; project administration, H.Z. All authors have read and agreed to the published version of the manuscript.

**Funding:** This work was funded by scientific research project of ecological restoration of Baopoling mountain in Sanya, China and a startup fund from Hainan University (KYQD (ZR) 1876).

**Acknowledgments:** The authors thank Liang Cong's assistance with the field experiment.

**Conflicts of Interest:** The authors declare no conflict of interest.

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
