# Peer review of "Reforestation Based on Mono-Plantation of Fast-Growing Tree Species Make It Difficult to Maintain (High) Soil Water Content in Tropics, a Case Study in Hainan Island, China"

_water, doi:10.3390/w12113077_

Round 1

Reviewer 1 Report

This paper presents the results of the impact of forestation with fast-growing non-native species on soil water content in a tropical environment. The study of the influence exerted by the increase of vegetal cover on water balance has gained the interest of researchers because of the impact on the availability of water resources.

The paper has, therefore, a potential interest for the readers of the journal, the results yet not novel provide additional confirmation of the findings of other authors that have already been  published in the scientific literature, some of them are reflected in the references included in the manuscript. The experimental layout and the methods used are correct so the conclusions are supported by the data.

Before accepting the manuscript for publication I would like suggest the authors addressing some minor issues in a new revised version:

  • Two of the key words are already included in the title so they should be deleted.
  • The text in lines 48-50 is the same as that in lines 41-43, please avoid repetition
  • The statement about “nearly no study has reported” that mono-specific, fast-growing plantations reduce the soil water content (line69) is not right. The impact of this type of forestation on water resources is largely well documented and reported in the scientific literature. Please provide with additional references on this.
  • Lines 71-85 would be better placed in the section on material and methods, where the study site is described
  • A more detailed description of the methods used  is desirable as for example (i) did the soil samples averaged (homogenized) for the whole depth(0-100 cm; line 143); (ii)it seems to exist some inconsistencies in the number of trees and leaves sampled as described in the manuscript and in the supplementary material, please clarify how many trees and leaves were sampled for each of the physiological traits; (iii) you should explain the statistical test and significant level used in section 2.3 and no later.
  • Line 165: “while, in the dry season, non-native species have much higher (6 times) transpiration rate, photosynthesis rate, stomatal conductance, leaf hydraulic conductivity, and 1eaf turgor loss point than those for native species (Figure 4).” What do you want to say with while? Please clarify. I strongly recommend a major editing work of all the manuscript. Sometimes it is hard to follow the thread of reasoning, especially in section 3 on results.
  • Line 212 Thus, native tree species’ appropriated hydraulic response to seasonal drought can maintain invariable high soil water content from wet to dry season “ Yet small the differences shown in figure 5 are statistically different so you cannot say they maintain invariable

Author Response

This paper presents the results of the impact of forestation with fast-growing non-native species on soil water content in a tropical environment. The study of the influence exerted by the increase of vegetal cover on water balance has gained the interest of researchers because of the impact on the availability of water resources.

The paper has, therefore, a potential interest for the readers of the journal, the results yet not novel provide additional confirmation of the findings of other authors that have already been  published in the scientific literature, some of them are reflected in the references included in the manuscript. The experimental layout and the methods used are correct so the conclusions are supported by the data.

Before accepting the manuscript for publication I would like suggest the authors addressing some minor issues in a new revised version:

Response: Thanks for your compliments and provided useful comments for improving our manuscript.

Two of the key words are already included in the title so they should be deleted.

Response: Corrected as suggested in line 33, page 1.

The text in lines 48-50 is the same as that in lines 41-43, please avoid repetition

Response: As suggested we have deleted redundant sentences and content to avoid repetition throughout the whole manuscript.

The statement about “nearly no study has reported” that mono-specific, fast-growing plantations reduce the soil water content (line69) is not right. The impact of this type of forestation on water resources is largely well documented and reported in the scientific literature. Please provide with additional references on this.

Response: As suggested we have changed this sentence into "Thus, it may be difficult to recover soil water content by reforestation using mono-planted fast-growing species".  

Lines 71-85 would be better placed in the section on material and methods, where the study site is described

Response: As per your recommendation, we have moved these parts to Material and Methods Section in currently revised manuscript.

A more detailed description of the methods used  is desirable as for example (i) did the soil samples averaged (homogenized) for the whole depth(0-100 cm; line 143);

 (ii)it seems to exist some inconsistencies in the number of trees and leaves sampled as described in the manuscript and in the supplementary material, please clarify how many trees and leaves were sampled for each of the physiological traits; (iii) you should explain the statistical test and significant level used in section 2.3 and no later.

Response: As suggested we have provided all your required information both soil samples and statistic test in lines 151-157, page 5. Namely: Every soil sample was homogenized for the whole depth (0-100 cm). Resampling was performed again in the dry season (February) in 2020. First, we used non-parameric test (generalized linear mixed effect model with poisson error family) (p<0.05) to test whether there were differences in transpiration related functional traits (transpiration rate, photosynthesis rate, stomatal conductance, leaf hydraulic conductivity and leaf turgor loss point) between the eight non-native and the eight native tropical tree species. We have also corrected inconsistent sampled numbers of trees and leaves in the Supplementary Material. 

Line 165: “while, in the dry season, non-native species have much higher (6 times) transpiration rate, photosynthesis rate, stomatal conductance, leaf hydraulic conductivity, and 1eaf turgor loss point than those for native species (Figure 4).” What do you want to say with while? Please clarify. I strongly recommend a major editing work of all the manuscript. Sometimes it is hard to follow the thread of reasoning, especially in section 3 on results.

Response: As suggested we have corrected this unclear sentence in lines 155-156,  page 6. We have also carefully revise any potential grammar errors and we hope current revised version has much improved now.  

Line 212 Thus, native tree species’ appropriated hydraulic response to seasonal drought can maintain invariable high soil water content from wet to dry season “ Yet small the differences shown in figure 5 are statistically different so you cannot say they maintain invariable

Response: As per your recommendation, we have corrected this incorrect description in lines 228-229, pages 8-9. Namely: thus, native tree species’ appropriated hydraulic response to seasonal drought could result in a slight decrease in the soil water content from the wet to the dry season.

Reviewer 2 Report

In this manuscript, “Reforestation based on mono-plantation of fast-growing tree species is difficult to recover soil water content in tropic, a case study in Hainan island, China”, the authors analyze how reforestation of an extremely degraded tropical rainforest with fast-growing tree species affects soil water levels. The results of this reforestation project are compared to an adjacent undisturbed tropical rainforest. The conclusions are based on a series of data collected from the trees and soil in both rainforests, the degraded and the undisturbed. A next step suggestion is briefly mentioned.

The manuscript provides an interesting analysis of the impact of non-native fast-growing tree species as a first step of reforestation on water soil. The results have potential use to improve reforestation in tropical rainforest, particularly from the water soil maintenance/recovery point of view. However, there are several areas where the manuscript needs improvement:

  1. A more detailed description of the reforestation project and the adjacent undisturbed rainforest would be beneficial.

The manuscript mentions that the degraded forest is a result of 20 years of limestone mining. It would be relevant to know which were the first measures implemented as part of the reforestation, in particular with regards to the topography and relief of the area. Where there any depressions that had to be filled in? If so, with which material or soil? Once the mining activity effects on the surface were restored, how is the project surface compared with the undisturbed forest? Are they both at the same height? Are there any depressions/mountains/valleys to take into consideration, due to their high influence on water soil levels? Where the two sites A and B chosen with the same topography characteristics?

  1. Further detail regarding water in the area is missing.

Is there any nearby city or village close to the analyzed sites? If so, where do their inhabitants get water from? Do they have a high demand for water? Which are the main sources of water demand in the area (agricultural use, industrial use, human consumption, etc.)? Is water ever scarce in the area? When? (dry/wet season).

  1. It would complement the research to identify an undisturbed forest site that is further away from the reforestation forest and analyze its soil water content. This would allow checking whether the soil water levels in Site B might have been reduced by the adjacent Site A. The lower soil water levels and the high water demand in Site A might have resulted in a absorption of soil water from the adjacent areas.

Would it be feasible to do this still? This would allow confirming the influence of the degraded forest over the surrounding area, and verify whether the difference between the regular soil water levels in an undisturbed forest and the reforested area is actually higher.

An alternative could be to compare with bibliography water soil levels of a similar tropical rainforest.

  1. Discussion – the title of the manuscript talks about the difficulty to recover soil water in the reforested area. However, there is an increase in the soil water in the reforested area between the dry and the wet season (the lost of water soil does not seem permanent). Why is this not considered enough recovery of the soil water? Which are the implications or negative effects of having these levels lower than in the undisturbed forest?

Also, given the recovery of the soil water from dry to wet season (even if the undisturbed forest levels are not reached), would it make sense to change the title to: ““Reforestation based on mono-plantation of fast-growing tree species make it difficult to maintain (high) soil water content in tropic, a case study in Hainan island, China”?

It would also be interesting to study whether soil water content increases and reaches the same level as in undisturbed forest once the reforested area moves to the next reforestation phase and starts incorporating native species. This would allow understanding whether the lower levels of soil water with the fast-growing species would continue in the future, or would be fully recovered to undisturbed forest levels once reforestation includes native species.

  1. Discussion – the manuscript focuses on the negative impact on soil water of fast-growing tree species, but it would be worth mentioning which are the benefits of these trees and to do a comparative analysis.

The manuscript mentions that there has been little impact due to typhoon and heavy rains in the past two years on the reforestation project, which is a benefit of the fast-growing trees. How beneficial would it be to not plant these fast-growing trees in the initial phase of the reforestation project? Is there any way in which both landslides could be avoided and soil water maintained? Are there any other benefits/impacts associated to the fast-growing trees at the beginning of reforestation that should be analyzed and considered?

  1. Conclusion

Do the authors still recommend that that fast-growing species are planted at the beginning of a reforestation project to avoid landslides, in spite of the negative impact on soil water?

A ‘next step’ is mentioned in the abstract and in the conclusion: mixing adjacent native tree species with the mono-plantation should be performed in the future. It would be necessary to provide more detail to this conclusion. How many years should the fast-growing trees grow alone in the reforestation project? Or, by which year should native tree species start to be planted together with the fast-growing species?

It would also be interesting to have a follow up research once native trees have been planted with the fast-growing trees in the reforestation project, in order to analyze whether and by when the soil water fully recovers to undisturbed forest levels.

What is intended with the last sentence, in lines 250-251? Does the team have plans to do some related research in the future? Do they recommend that similar research is implemented?

  1. Other issues
  • The text is quite duplicative. It would benefit from a revision to avoid duplicating the same information and the same long sentences over and over.
  • Figures 3 and 4 don’t show the legend in all the graphics, and it is very confusing. Recommend to use the legends and graphic patterns as used in Figure 5.
  • Lines 1, 7, 16: belong to the manuscript template, delete.
  • Line 33: the sentence is not clear, it needs to specify that the next step is to mix native-species with the current fast-growing non-native species. Otherwise, it can be understood as substituting all fast-growing species, instead of mixing them. Typo: ‘perform’.
  • Line 42, 49: not all plantations of commercial trees result in very high deforestation and degradation. There are sustainable forest management practices that avoid deforestation and degradation while allowing for the plantation of trees for commercial use. This is also the case of forests certified by a sustainable forest management scheme, such as the Forest Stewardship Council (FSC).
  • Lines 48-50: it is a duplication of line 42. Revise to avoid duplication.
  • Line 86: revise, hard to understand.
  • Lines 105, 111: is Figure 1 the right reference?
  • Line 137: should this reference be Figure 2?
  • Lines 239 – 240. The sentence is confusing and not understood.
  • Line 249: the use of the word ‘alter’ is not clear. What does this refer to?
  • Line 250: the use of the word ‘warrant’ is not clear. What does this refer to?

Author Response

In this manuscript, “Reforestation based on mono-plantation of fast-growing tree species is difficult to recover soil water content in tropic, a case study in Hainan island, China”, the authors analyze how reforestation of an extremely degraded tropical rainforest with fast-growing tree species affects soil water levels. The results of this reforestation project are compared to an adjacent undisturbed tropical rainforest. The conclusions are based on a series of data collected from the trees and soil in both rainforests, the degraded and the undisturbed. A next step suggestion is briefly mentioned.

The manuscript provides an interesting analysis of the impact of non-native fast-growing tree species as a first step of reforestation on water soil. The results have potential use to improve reforestation in tropical rainforest, particularly from the water soil maintenance/recovery point of view. However, there are several areas where the manuscript needs improvement:

A more detailed description of the reforestation project and the adjacent undisturbed rainforest would be beneficial.

Response: Thanks for your compliments and useful comments. We have followed your suggestions to provide all your required information in currently revised version.

The manuscript mentions that the degraded forest is a result of 20 years of limestone mining. It

would be relevant to know which were the first measures implemented as part of the reforestation, in particular with regards to the topography and relief of the area. Where there any depressions that had to be filled in? If so, with which material or soil? Once the mining activity effects on the surface were restored, how is the project surface compared with the undisturbed forest? Are they both at the same height? Are there any depressions/mountains/valleys to take into consideration, due to their high influence on water soil levels? Where the two sites A and B chosen with the same topography characteristics?

Response: As suggested, we have provided your required information on the reforestation project and the adjacent undisturbed rainforest in lines 128-134, page 3. Namely: Due to 20 years of limestone mining associated with the cement industry, this 0.2 km2 highly degraded tropical forest is now consisting merely of bare rocks that do not support plant life (Fig. 2). Areas of the BPL outside of this 0.2 km2 degraded area have been significantly disturbed, and therefore have remained as a species rich tropical rainforest (Fig.2). In May 2016, we used the adjacent undisturbed forest as a reference to perform a reforestation project in BPL with the aim to recover soil water content and vegetation cover of BPL. The slope and the deep soil layers of the undisturbed forest area was used as a reference to reconstruct slope and soil layers for the reforested area. Then, refilling of the area was performed with the help of the soil from the undisturbed tropical rainforest areas to monoculture seedlings (3 m height and 2 cm diameter at breast height (DBH)) of eight fast-growing tree species, Terminalia neotaliala, Bombax malabarica, Cleistanthus sumatranus, Ficus microcarpa, Muntingia colabura, Acacia mangium,  Leucaena glauca and Bougainvillea spectabilis. Seedlings of these eight fast-growing species were purchased commercially. These species are known to be fast-growing, and have high survival rates within the study region. Therefore, we reasoned that these eight species should have high potential to prevent landslide during frequent typhoon and heavy rains. These eight species were separately monocultured from the top to the bottom of BPL (Fig. 2) and planting density for each of the species was maintained at 100stems per hectare. The restoration project was finished at end of year 2016. In 2019,thirty plots, each of 20×20 m2 (an area of 400 m2 for each plot) that were at least 100-300 m apart from one another, were randomly sampled across the adjacent undisturbed old-growth forest. Within each plot, all freestanding trees with diameter of ≥ 1 cm at breast height (DBH) were measured and identified to species. We finally found 80 tree species in the undisturbed old-growth forest and we selected the 8 tree species (200-300 stems per hectare) (Bridelia tomentosa, Radermachera frondosa, Lepisanthes rubiginosa, Rhaphiolepis indica, Pterospermum heterophyllum, Fissistigma oldhamii, Psychotria rubra and Cudraniaco chinchinensis) as our candidate dominant slow-growing tree species.

Further detail regarding water in the area is missing.

Is there any nearby city or village close to the analyzed sites? If so, where do their inhabitants get water from? Do they have a high demand for water? Which are the main sources of water demand in the area (agricultural use, industrial use, human consumption, etc.)? Is water ever scarce in the area? When? (dry/wet season).

Response: As per your recommendation, we have provided this information in lines 90-93,  page 3. Namely: The inhabitants of the village near BPL get their water supply from the nearby pond and the water works in Sanya city. Additionally, the occurrence of a major cement factory (Huasheng cement factory, China) near BPL exponentially increases the water demand so that the city sometimes would get very limited water supply in the dry season.

It would complement the research to identify an undisturbed forest site that is further away from the reforestation forest and analyze its soil water content. This would allow checking whether the soil water levels in Site B might have been reduced by the adjacent Site A. The lower soil water levels and the high water demand in Site A might have resulted in a absorption of soil water from the adjacent areas. Would it be feasible to do this still? This would allow confirming the influence of the degraded forest over the surrounding area, and verify whether the difference between the regular soil water levels in an undisturbed forest and the reforested area is actually higher.

Response: Thanks for this useful suggestion and we think it may be not feasible to find out a new undisturbed forest that is further away from the reforestation forest or to perform this comparative research. That is because the reforestation project use the slope and the deep soil layer for the undisturbed tropical forest as a reference to reconstruct slope and soil layers. Then refilling the same soil from the undisturbed tropical rainforest to plant tree species. Thus the reforestation project and the undisturbed tropical rainforest have the same topographic and soil environments. However topographic and soil environments in another undisturbed forest that is further away from the reforestation forest should be totally different and we cannot guarantee the final differences in soil water content between reforestation forest and the new undisturbed forest that is further away from the reforestation forest can be attributed to mono-planting fast-growing tree species.  

An alternative could be to compare with bibliography water soil levels of a similar tropical rainforest.

Response: Thanks for another useful suggestion, but comparing with bibliography water soil levels of a similar tropical rainforest should require this similar tropical rainforest have the same topographic and soil environments. Since this reforestation project used the adjacent undisturbed forest as a reference to reconstruct the same topographic and soil environments, adjacent undisturbed forest should be the unique best choice.

Discussion – the title of the manuscript talks about the difficulty to recover soil water in the reforested area. However, there is an increase in the soil water in the reforested area between the dry and the wet season (the lost of water soil does not seem permanent). Why is this not considered enough recovery of the soil water? Which are the implications or negative effects of having these levels lower than in the undisturbed forest? Also, given the recovery of the soil water from dry to wet season (even if the undisturbed forest levels are not reached), would it make sense to change the title to: ““Reforestation based on mono-plantation of fast-growing tree species make it difficult to maintain (high) soil water content in tropic, a case study in Hainan island, China”?

Response: We agree, so we have used your suggested new title in currently revised version.

It would also be interesting to study whether soil water content increases and reaches the same level as in undisturbed forest once the reforested area moves to the next reforestation phase and starts incorporating native species. This would allow understanding whether the lower levels of soil water with the fast-growing species would continue in the future, or would be fully recovered to undisturbed forest levels once reforestation includes native species.

Response: As suggested, we have pointed out this key point in lines 245-247, page 9. 

Discussion – the manuscript focuses on the negative impact on soil water of fast-growing tree species, but it would be worth mentioning which are the benefits of these trees and to do a comparative analysis. The manuscript mentions that there has been little impact due to typhoon and heavy rains in the past two years on the reforestation project, which is a benefit of the fast-growing trees. How beneficial would it be to not plant these fast-growing trees in the initial phase of the reforestation project? Is there any way in which both landslides could be avoided and soil water maintained? Are there any other benefits/impacts associated to the fast-growing trees at the beginning of reforestation that should be analyzed and considered?

Response: To be best of our knowledge mono-planting fast-growing species with high survival rate is the only strategy to prevent landslide, so we still suggest to use this way to perform reforestation. As per your recommendation, we have provided other beneficial got from mono-planting non-native fast-growing trees at the beginning of reforestation in lines 274-283, pages 9.

Conclusion

Do the authors still recommend that fast-growing species are planted at the beginning of a reforestation project to avoid landslides, in spite of the negative impact on soil water?

Response: Yes, we still recommend that fast-growing species are planted at the beginning of a reforestation project to avoid landslides, in spite of the negative impact on soil water. We have also pointed out this point in lines 234-243, page 9.

A ‘next step’ is mentioned in the abstract and in the conclusion: mixing adjacent native tree species with the mono-plantation should be performed in the future. It would be necessary to provide more detail to this conclusion. How many years should the fast-growing trees grow alone in the reforestation project? Or, by which year should native tree species start to be planted together with the fast-growing species? It would also be interesting to have a follow up research once native trees have been planted with the fast-growing trees in the reforestation project, in order to analyze whether and by when the soil water fully recovers to undisturbed forest levels.

Response: Thanks very much for these useful comments and we have also pointed out in lines 245-247, page 9, these two future key research questions.

What is intended with the last sentence, in lines 250-251? Does the team have plans to do some related research in the future? Do they recommend that similar research is implemented?

Response: We intend to convey our reforestation can provide some useful clue for performing reforestation to recover other degraded tropical rainforests. However, we have realized that this can be fulfilled when performing your suggested future researches to improve our reforestation project. Thus we have deleted this sentence in currently revised version.

Other issues

The text is quite duplicative. It would benefit from a revision to avoid duplicating the same information and the same long sentences over and over.

Response: As per your recommendation, we have carefully revised the whole manuscript to avoid duplicating the same information and the same long sentences over and over. We hope currently revised version has much improved now.

Figures 3 and 4 don’t show the legend in all the graphics, and it is very confusing. Recommend to use the legends and graphic patterns as used in Figure 5.

Lines 1, 7, 16: belong to the manuscript template, delete.

Response: Deleted as suggested.

Line 33: the sentence is not clear, it needs to specify that the next step is to mix native-species with the current fast-growing non-native species. Otherwise, it can be understood as substituting all fast-growing species, instead of mixing them. Typo: ‘perform’.

Response: Corrected as suggested in line]31, page 1.

Line 42, 49: not all plantations of commercial trees result in very high deforestation and degradation. There are sustainable forest management practices that avoid deforestation and degradation while allowing for the plantation of trees for commercial use. This is also the case of forests certified by a sustainable forest management scheme, such as the Forest Stewardship Council (FSC).

Response: We agree, so we have changed to unreasonable land use but not plantation of commercial trees.  

Lines 48-50: it is a duplication of line 42. Revise to avoid duplication.

Response: As suggested, we have deleted responding duplicated sentences.  

Line 86: revise, hard to understand.

Response: Corrected as suggested.

Lines 105, 111: is Figure 1 the right reference?

Response: No, and corrected as suggested in line 98, page 3.

Line 137: should this reference be Figure 2?

Response: Yes, and corrected as suggested in line 108, page 2.

Lines 239 – 240. The sentence is confusing and not understood.

Response: Corrected.

Line 249: the use of the word ‘alter’ is not clear. What does this refer to?

Response: We'd apologized for this unclear sentence and we have rewritten this sentence.

Line 250: the use of the word ‘warrant’ is not clear. What does this refer to?

Response: Sorry for this unclear sentence and we have rewritten this sentence.

Reviewer 3 Report

This article is very important nowadays. It deals with current problems related to the level of groundwater.
The authors investigated the effect of reforestation on increasing retention.
They took into account the monocultures that are necessary to protect the soil from erosion. The variable climatic conditions (dry and rainy seasons) make it difficult to take care of the amount of water in the soil. The reason is that plants (8 fast-growing species) take up just as much water all year round - and this results in a lot of drainage in the dry season.
The authors also show the direction and guidelines for further research.
In my opinion, the article is worth publishing in "Water"

Detailed comments;
1) the title is long but reflects the content of the article
2) the abstract contains the research subject and correctly informs about the obtained results
3) Keywords - remove those that are repeated in the title. Some others can be added
4) presentation of the research problem well developed, based on the current literature. Graphical hypothesis, which makes it very easy to understand the role of trees in the water cycle.
5) the methodology is quite briefly described in the test of the article, but more extensively in supplementary materials - which is quite a good solution
6) Very succinct results (maybe a bit too little described) - but the charts are clear and I think there is no need to extend this chapter.
7) Discussion - supported by literature
refers to the results obtained
8) L; 259 - remove the dot
9) L; 263-269 - to be removed
10) I don't see  inf about supplementary matterials at the end

Author Response

This article is very important nowadays. It deals with current problems related to the level of groundwater. The authors investigated the effect of reforestation on increasing retention.
They took into account the monocultures that are necessary to protect the soil from erosion. The variable climatic conditions (dry and rainy seasons) make it difficult to take care of the amount of water in the soil. The reason is that plants (8 fast-growing species) take up just as much water all year round - and this results in a lot of drainage in the dry season.The authors also show the direction and guidelines for further research.In my opinion, the article is worth publishing in "Water".

Response: Thanks for your positive comment.

Detailed comments;
1) the title is long but reflects the content of the article

Response: Thanks for your positive comment.

2) the abstract contains the research subject and correctly informs about the obtained results

Response: Thanks for your positive comment.

3) Keywords - remove those that are repeated in the title. Some others can be added

Response: Corrected as suggested in lines33, page 1.

4) presentation of the research problem well developed, based on the current literature. Graphical hypothesis, which makes it very easy to understand the role of trees in the water cycle.

Response: Thanks for your positive comment.

5) the methodology is quite briefly described in the test of the article, but more extensively in supplementary materials - which is quite a good solution

Response: Thanks for your positive comment.

6) Very succinct results (maybe a bit too little described) - but the charts are clear and I think there is no need to extend this chapter.

Response: Thanks for your positive comment.

7) Discussion - supported by literature refers to the results obtained

Response: Thanks for your positive comment.

8) L; 259 - remove the dot

Response: Corrected as suggested in lines page.

9) L; 263-269 - to be removed

Response: Deleted as suggested.

10) I don't see inf about supplementary materials at the end

Response: As per your recommendation, we have provided what supplementary materials pointed in line 146-147, page 6.